# YaRN: Efficient Context Window Extension of Large Language Models

**Bowen Peng**[1]  **Jeffrey Quesnelle**[1]  **Honglu Fan**[23]  **Enrico Shippole**

[1]Nous Research  [2]EleutherAI  [3]University of Geneva

## Abstract

Rotary Position Embeddings (RoPE) have been shown to effectively encode positional information in transformer-based language models. However, these models fail to generalize past the sequence length they were trained on. We present YaRN (Yet another RoPE extensioN method), a compute-efficient method to extend the context window of such models, requiring 10x less tokens and 2.5x less training steps than previous methods. Using YaRN, we show that LLaMA models can effectively utilize and extrapolate to context lengths much longer than their original pre-training would allow, while also surpassing previous the state-of-the-art at context window extension. In addition, we demonstrate that YaRN exhibits the capability to extrapolate beyond the limited context of a fine-tuning dataset. The models fine-tuned using YaRN has been made available and reproduced online up to 128k context length.

## 1 Introduction

Transformer-based Large Language Models(Vaswani et al., 2017) (LLMs) have become the near-ubiquitous choice for many natural language processing (NLP) tasks where long-range abilities such as *in-context learning* (ICL) has been crucial. In performing the NLP tasks, the maximal length of the sequences (the *context window*) determined by its training processes has been one of the major limits of a pretrained LLM. Being able to dynamically extend the context window via a small amount of fine-tuning (or without fine-tuning) has become more and more desirable. To this end, the position encodings of transformers are the center of the discussions.

The original Transformer architecture used an absolute sinusoidal position encoding, which was later improved to a learnable absolute position encoding (Gehring et al., 2017). Since then, relative positional encoding schemes (Shaw et al., 2018) have further increased the performance of Transformers. Currently, the most popular relative positional encodings are *T5 Relative Bias* (Roberts et al., 2019), *RoPE* (Su et al., 2022), *XPos* (Sun et al., 2022), and *ALiBi* (Press et al., 2022).

One reoccurring limitation with positional encodings is the inability to generalize past the context window seen during training. While some methods such as ALiBi are able to do limited generalization, none are able to generalize to sequences significantly longer than their pre-trained length (Kazemnejad et al., 2023).

Some works have been done to overcome such limitation. (Chen et al., 2023) and concurrently (kaiokendev, 2023) proposed to extend the context length by slightly modifying RoPE via Position Interpolation (PI) and fine-tuning on a small amount of data. As an alternative, (bloc97, 2023a) proposed the "NTK-aware" interpolation by taking the loss of high frequency into account. Since then, two improvements of the "NTK-aware" interpolation have been proposed, with different emphasis:

- the "Dynamic NTK" interpolation method (emozilla, 2023) for pre-trained models without fine-tuning.
- the "NTK-by-parts" interpolation method (bloc97, 2023b) which performs the best when fine-tuned on a small amount of longer-context data.

The "NTK-aware" interpolation and the "Dynamic NTK" interpolation have already seen their presence in the open-source models such as Code Llama (Rozière et al., 2023) (using "NTK-aware" interpolation) and Qwen 7B (qwe) (using "Dynamic NTK").

In this paper, in addition to making a complete account of the previous unpublished works on the "NTK-aware", the "Dynamic NTK" and the "NTK-by-parts" interpolations, we present YaRN (Yet another RoPE extensioN method), an improved method to efficiently extend the context window of models trained with Rotary Position Embeddings (RoPE) including the LLaMA (Touvron et al., 2023a), the GPT-NeoX (Black et al., 2022), and the PaLM (Chowdhery et al., 2022) families of models.

The relationship between different methods and how they evolve into YaRN can be summarized into the following diagram:

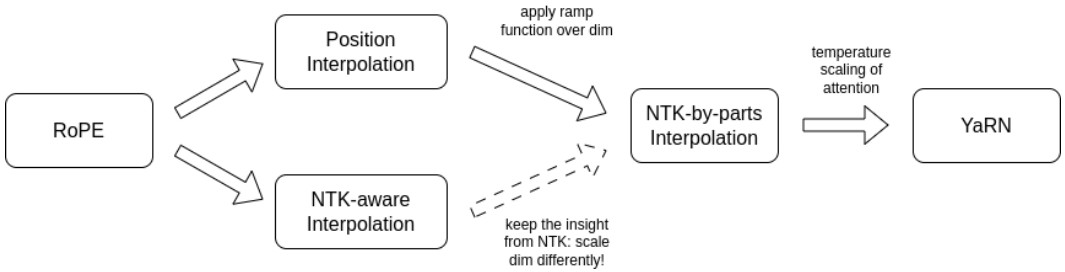

Figure 1: An outline of the relationship between different interpolation methods.

YaRN reaches state-of-the-art performances in context window extensions after fine-tuning on less than ∼0.1% of the original pre-training data. In the meantime, by combining with the inference-time technique called Dynamic Scaling, the Dynamic-YaRN allows for more than 2x context window extension without any fine-tuning.

## 2 BACKGROUND AND RELATED WORK

### 2.1 ROTARY POSITION EMBEDDINGS

The basis of our work is the Rotary Position Embedding (RoPE) introduced in (Su et al., 2022). We work on a hidden layer where the set of hidden neurons are denoted by $D$. Given a sequence of vectors $\boldsymbol{x}_1, \cdots, \boldsymbol{x}_L \in \mathbb{R}^{|D|}$, following the notation of (Su et al., 2022), the attention layer first converts the vectors into the query vectors and the key vectors:

$$\boldsymbol{q}_m = f_q(\boldsymbol{x}_m, m) \in \mathbb{R}^{|D|}, \ \boldsymbol{k}_n = f_k(\boldsymbol{x}_n, n) \in \mathbb{R}^{|D|}. \tag{1}$$

Next, the attention weights are calculated as

$$\text{softmax}(\frac{\boldsymbol{q}_m^T \boldsymbol{k}_n}{\sqrt{|D|}}), \tag{2}$$

where $\boldsymbol{q}_m, \boldsymbol{k}_n$ are considered as column vectors so that $\boldsymbol{q}_m^T \boldsymbol{k}_n$ is simply the Euclidean inner product. In RoPE, we first assume that $|D|$ is even and identify the embedding space and the hidden states as complex vector spaces:

$$\mathbb{R}^{|D|} \cong \mathbb{C}^{|D|/2}$$

where the inner product $\boldsymbol{q}^T \boldsymbol{k}$ becomes the real part of the standard Hermitian inner product $\text{Re}(\boldsymbol{q}^* \boldsymbol{k})$. More specifically, the isomorphisms interleave the real part and the complex part

$$\big((\boldsymbol{x}_m)_1, \cdots, (\boldsymbol{x}_m)_{|D|}\big) \mapsto \big((\boldsymbol{x}_m)_1 + i(\boldsymbol{x}_m)_2, \cdots, ((\boldsymbol{x}_m)_{|D|-1} + i(\boldsymbol{x}_m)_{|D|})\big), \tag{3}$$

$$\big((\mathbf{q}_m)_1, \cdots, (\mathbf{q}_m)_{|D|}\big) \mapsto \big((\mathbf{q}_m)_1 + i(\mathbf{q}_m)_2, \cdots, ((\mathbf{q}_m)_{|D|-1} + i(\mathbf{q}_m)_{|D|})\big). \tag{4}$$

To convert embeddings $\boldsymbol{x}_m, \boldsymbol{x}_n$ into query and key vectors, we are first given $\mathbb{R}$-linear operators

$$\boldsymbol{W}_q, \boldsymbol{W}_k : \mathbb{R}^{|D|} \to \mathbb{R}^{|D|}.$$

Let $\boldsymbol{\theta} = \text{diag}(\theta_1, \cdots, \theta_{|D|/2})$. In complex coordinates, we define

$$f_{\boldsymbol{W}}(\boldsymbol{x}_m, m, \boldsymbol{\theta}) = e^{im\boldsymbol{\theta}} \boldsymbol{W} \boldsymbol{x}_m, \tag{5}$$

for any linear operator $\boldsymbol{W}$. The functions $f_q$, $f_k$ in RoPE are given by

$$f_q = f_{\boldsymbol{W}_q}, \ f_k = f_{\boldsymbol{W}_k}. \tag{6}$$

where $\theta_d = b^{-2d/|D|}$ and $b = 10000$. This way, RoPE associates each (complex-valued) hidden neuron with a separate frequency $\theta_d$. The benefit of doing so is that the dot product between the query vector and the key vector only depends on the relative distance $m - n$.

In later discussions, a context length interpolation usually aims to modify the equation Eq. 5. To set up a uniform convention for these discussions, note that a modification $f'_{\boldsymbol{W}}$ can take the following form:

$$f'_{\boldsymbol{W}}(\boldsymbol{x}_m, m, \boldsymbol{\theta}) = f_{\boldsymbol{W}}(\boldsymbol{x}_m, g(m), \boldsymbol{h}(\boldsymbol{\theta})), \tag{7}$$

where $g(m)$ is a map between real numbers and $\boldsymbol{h}(\boldsymbol{\theta})$ acts on the entries of the diagonal matrix $\boldsymbol{\theta}$ uniformly by $\text{diag}(h(\theta_1), \cdots, h(\theta_{|D|/2}))$ according to a function $h$. $g$ and $h$ are method-dependent functions.

In the subsequent sections, when we introduce a new interpolation method of the form Eq. 7, we only specify the functions $g(m)$ and $h(\theta_d)$.

## 2.2 Additional notations

Given the pretrained maximal context length $L$, our goal is to extend it to $L' > L$ either with or without finetuning. We introduce the notion of *scale factor s* defined by $s = \frac{L'}{L}$.

For the convenience of some discussions, we also introduce *wavelength* $\lambda_d$ associated with the $d$-th hidden dimension of RoPE as follows:

$$\lambda_d = \frac{2\pi}{\theta_d} = 2\pi b^{\frac{2d}{|D|}}. \tag{8}$$

The wavelength describes the length of tokens needed in order for the rotary position embedding at dimension $d$ to perform a full rotation ($2\pi$).

## 2.3 Related work

Position Interpolation (PI) is one of the earlier works extending context lengths of RoPE proposed by Chen et al. (2023), and concurrently kaiokendev (2023). Under the notation of Eq. 7, it is setting

$$g(m) = s \cdot m, \ \boldsymbol{h}(\boldsymbol{\theta}) = \boldsymbol{\theta}, \tag{9}$$

where $s$ is the scale factor $\frac{L'}{L}$. We include some details in Appendix A.1.

ReRoPE (Su, 2023) also aims to extend the context size of existing models pre-trained with RoPE, and claims "infinite" context length without needing any fine-tuning. This claim is backed by a monotonically decreasing loss with increasing context length up to 16k on the Llama 2 13B model. It achieves context extension by modifying the attention mechanism and thus is not purely an embedding interpolation method. Since it is currently not compatible with Flash Attention 2 (Dao, 2023) and requires two attention passes during inference, we do not consider it for comparison.

Concurrently with our work, LM-Infinite (Han et al., 2023) proposes similar ideas to YaRN, but focuses on "on-the-fly" length generalization for non-fine-tuned models. Since they also modify the attention mechanism of the models, it is not an embedding interpolation method and is not immediately compatible with Flash Attention 2.

## 3 Methodology

Whereas PI stretches all RoPE dimensions equally, we find that the theoretical interpolation bound described by PI (Chen et al., 2023) is insufficient at predicting the complex dynamics between RoPE and the LLM's internal embeddings. In the following subsections, we describe the main issues with PI we have individually identified and solved, so as to give the readers the context, origin and justifications of each method which we use in concert to obtain the full YaRN method.

### 3.1 Loss of High Frequency information - "NTK-aware" interpolation

If we look at rotary position embeddings (RoPE) only from an information encoding perspective, it was shown in (Tancik et al., 2020), using Neural Tangent Kernel (NTK) theory, that deep neural networks have trouble learning high frequency information if the input dimension is low and the corresponding embeddings lack high frequency components. Here we can see the similarities: a token's positional information is one-dimensional, and RoPE expands it to an n-dimensional complex vector embedding. RoPE closely resembles Fourier Features (Tancik et al., 2020) in many aspects, as it is possible to define RoPE as a special 1D case of a Fourier Feature.

In the case of Positional Interpolation (PI), as we strech all dimensions equally by a factor $s$, it removes the high frequency components of RoPE. This degradation is worsened as the scaling factor $s$ grows, and at some point, the network will not be able to recover. Previous fine-tunes (kaiokendev, 2023) (Chen et al., 2023) (Together.ai, 2023) (Quesnelle et al., 2023) using PI were only able to achieve a scaling factor of roughly $s = 8$ before the LLM's outputs starts to degrade, even after fine-tuning.

In order to alleviate this issue, the "NTK-aware" interpolation was developed in (bloc97, 2023a). Instead of scaling every dimension of RoPE equally by a factor $s$, we spread out the interpolation pressure across multiple dimensions by scaling high frequencies less and low frequencies more. One can obtain such a transformation in many ways, but the simplest would be to perform a base change on the value of $\theta$. The details are described in the Appendix A.2 and the method has seen some open-source adoptions[1].

One main issue of this "NTK-aware" scaling is that it is very difficult to determine what optimal base should be used for an intended context extension by $s$ times. The best base to use for "NTK-aware" interpolation usually has to be found empirically, which significantly increases the difficulty and cost of obtaining a successful fine-tuned model. Despite its limitations, the observations from the NTK theory is valid and the following idea is still maintained and executed in a different way in the "NTK-by-parts" interpolation introduced in the next section.

### 3.2 Loss of Relative Local Distances - "NTK-by-parts" interpolation

To understand why "NTK-aware" interpolation works better than PI and to eliminate its disadvantages, we have to take a closer look at RoPE. In this section, we think heavily in terms of the wavelengths $\lambda_d$ defined in Eq. 8 in the formula of RoPE. For simplicity, we omit the subscript $d$ in $\lambda_d$ and the reader is encouraged to think about $\lambda$ as the wavelength of an arbitrary periodic function.

In theory, as RoPE is a relative position embedding, it should be quite surprising that it fails to generalize to unseen longer context sizes. However, we can show that in practice, RoPE does not only encode relative position. One observation we can make is that given a context size $L$, there are some dimensions $d$ where the wavelength is longer than the maximum context length seen during pretraining ($\lambda > L$), this suggests that some dimensions' rotary embeddings might not be distributed evenly in the rotational domain (i.e. does not perform a full rotation for the entire training context size). In such cases, we presume having unique position pairs[2] implies that the absolute positional information remains intact in those dimensions. On the contrary, when the wavelength is short, only relative positional information is accessible to the network.

Given these observations, we can see that it is important to not touch the dimensions that only encode relative positional information, as they are crucial for the network to distinguish the relative order of nearby tokens. Meanwhile, dimensions that only encode absolute positional information should always be interpolated, as larger distances will be out of distribution. Instead of arbitrarily changing the base in "NTK-aware" interpolation (which basically does something similar to what is described here), we can formulate an explicit and targeted interpolation method that takes in account all of the above.

---

[1]We note that shortly before the release of this article, Code Llama (Rozière et al., 2023) was released and uses "NTK-aware" scaling by manually scaling the base $b$ to 1M, in which they call this method as RoPE "adjusted base frequency" (ABF).

[2]Since the dimension never rotates fully at least once during pre-training, if we pick the first token as the anchor, every other token during pre-training has an unique distance to it, which the neural network can use to determine its absolute position.

In other words,

- if the wavelength $\lambda$ is much smaller than the context size $L$, we do not interpolate;
- if the wavelength $\lambda$ is equal to or bigger than the context size $L$, we want to only interpolate and avoid any extrapolation (unlike the previous "NTK-aware" method);
- dimensions in-between can have a bit of both, similar to the "NTK-aware" interpolation.

As a result, it is more convenient to introduce the ratio $r = \frac{L}{\lambda}$ between the original context size $L$ and the wavelength $\lambda$. This ratio represents the number of rotations a certain RoPE dimension makes given a fixed pretrained context length $L$. In the $d$-th hidden state, the ratio $r$ depends on $d$ in the following way:

$$r(d) = \frac{L}{\lambda_d} = \frac{L}{2\pi b^{\frac{2d}{|D|}}}.$$ (10)

In order to define the boundary of the different interpolation strategies as above, we introduce two extra parameters $\alpha, \beta$. All hidden dimensions $d$ where $r(d) < \alpha$ are those where we linearly interpolate by a scale $s$ (exactly like PI, avoiding any extrapolation), and the $d$ where $r(d) > \beta$ are those where we do not interpolate at all. Define the ramp function $\gamma$ to be

$$\gamma(r) = \begin{cases} 0, & \text{if } r < \alpha \\ 1, & \text{if } r > \beta \\ \dfrac{r - \alpha}{\beta - \alpha}, & \text{otherwise.} \end{cases}$$ (11)

With the help of the ramp function, the "NTK-by-parts" method can be described as follows.

**Definition 1** *The "NTK-by-parts" interpolation is a modification of RoPE using Eq. 7 with the following functions*[3].

$$g(m) = m$$ (12)

$$h(\theta_d) = \left(1 - \gamma\big(r(d)\big)\right)\frac{\theta_d}{s} + \gamma\big(r(d)\big)\theta_d.$$ (13)

The values of $\alpha$ and $\beta$ should be tuned on a case-by-case basis. For example, we have found experimentally that for the Llama family of models, good values for $\alpha$ and $\beta$ are $\alpha = 1$ and $\beta = 32$.

Using the techniques described in this section, a variant of the resulting method was released under the name "NTK-by-parts" interpolation (bloc97, 2023b). This improved method performs better than the previous PI (Chen et al., 2023) and "NTK-aware" 3.1 interpolation methods, both with non-fine-tuned models and with fine-tuned models, as shown in (bloc97, 2023b) and Section 4.2.

### 3.3 YARN

In addition to the previous interpolation techniques, we also observe that introducing a temperature $t$ on the logits before the attention softmax has a uniform impact on perplexity regardless of the data sample and the token position over the extended context window (See Appendix A.3). More precisely, instead of Eq. 2, we modify the computation of attention weights into

$$\text{softmax}\left(\frac{\boldsymbol{q}_m^T \boldsymbol{k}_n}{t\sqrt{|D|}}\right).$$ (14)

The reparametrization of RoPE as a set of 2D matrices has a clear benefit on the implementation of this attention scaling: we can instead use a "length scaling" trick which scales both $\boldsymbol{q}_m$ and $\boldsymbol{k}_n$ by a constant factor $\sqrt{1/t}$ by simply scaling the complex rotary position embeddings by the same amount. With this, YaRN can effectively alter the attention mechanism without modifying its code. Furthermore, it has zero overhead during both inference and training, as rotary position embeddings are generated in advance and are reused for all forward passes. Combining it with the "NTK-by-parts" interpolation, we have the YaRN method.

---

[3]The interpolation by linear ramp on $h$ may have alternatives, such as a harmonic mean over $\theta_d/s$ and $\theta_d$ converted from a linear interpolation on wavelengths. The choice of $h$ here was for the simplicity of implementation, but both would work.

**Definition 2** *By the "YaRN method", we refer to a combination of the attention scaling in Eq. 14 and the "NTK-by-parts" interpolation introduced in Section 3.2.*

For LLaMA and Llama 2 models, we recommend the following values:

$$\sqrt{\frac{1}{t}} = 0.1 \ln(s) + 1. \tag{15}$$

The equation above is found by fitting $\sqrt{1/t}$ at the lowest perplexity against the scale extension by various factors $s$ using the "NTK-by-parts" method (Section 3.2) on LLaMA 7b, 13b, 33b and 65b models without fine-tuning. We note that the same values of $t$ also apply fairly well to Llama 2 models (7b, 13b and 70b). It suggests that the property of increased entropy and the temperature constant $t$ may have certain degree of "universality" and may be generalizable across some models and training data.

The YaRN method combines all our findings and surpasses all previous methods in both fine-tuned and non-fine-tuned scenarios. Thanks to its low footprint, YaRN allows for direct compatibility with libraries that modify the attention mechanism such as Flash Attention 2 (Dao, 2023).

### 3.4 Dynamic Scaling - "Dynamic NTK" interpolation

In a lot of use cases, multiple forward-passes are performed with varying sequence lengths from 1 to the maximal context size. A typical example is the autoregressive generation where the sequence lengths increment by 1 after each step. There are two ways of applying an interpolation method that uses a scale factor $s$ (including PI, "NTK-aware", "NTK-by-parts" and YaRN):

1. Throughout the whole inference cycle, the embedding layer is fixed including the scale factor $s = L'/L$ where $L'$ is the fixed number of extended context size.
2. In each forward-pass, the position embedding updates the scale factor $s = \max(1, l'/L)$ where $l'$ is the sequence length of the current sequence.

The problem of (1) is that the model may experience a performance discount at a length less than $L$ and an abrupt degradation when the sequence length is longer than $L'$. But by doing Dynamic Scaling as (2), it allows the model to gracefully degrade instead of immediately breaking when hitting the trained context limit $L'$. We call this inference-time method the Dynamic Scaling method. When it is combined with "NTK-aware" interpolation, we call it "Dynamic NTK" interpolation. It first appeared in public as a reddit post in (emozilla, 2023).

One notable fact is that the "Dynamic NTK" interpolation works exceptionally well on models pre-trained on $L$ without any finetuning ($L' = L$). This is supported by the experiment in Appendix B.7.

Often in the repeated forward-passes, the kv-caching (Chen, 2022) is applied so that we can reuse the previous key-value vectors and improve the overall efficiency. We point out that in some implementations when the rotary position embeddings are cached, some care has to be taken in order to modify it for Dynamic Scaling with kv-caching. The correct implementation should cache the kv-embeddings before applying rotary position embeddings, as the RoPE of every token changes when $s$ changes.

## 4 Experiments

### 4.1 Training

We broadly followed the training and evaluation procedures as outlined in Chen et al. (2023).

For training the 128k context window size models, we extended the Llama 2 (Touvron et al., 2023b) 7B and 13B parameter models. No changes were made to the LLaMA model architecture other than the calculation of the embedding frequencies as described in Section 3.3 with $s = 16$ and $s = 32$.

We used a learning rate of $2 \times 10^{-5}$ with no weight decay and a linear warmup of 20 steps along with AdamW (Loshchilov and Hutter, 2019) $\beta_1 = 0.9$ and $\beta_2 = 0.95$. For the $s = 16$ model, we

fine-tuned for 400 steps with global batch size $64$ using PyTorch (Paszke et al., 2019) Fully Sharded Data Parallelism (Zhao et al., 2023) and Flash Attention 2 (Dao, 2023) on the PG19 dataset (Rae et al., 2020) chunked into 64k segments bookended with the BOS and EOS token. For $s = 32$ we followed the same procedure, but due to compute constraints, we started from the finished $s = 16$ checkpoint and trained for only an additional 200 steps. Note that the $s = 32$ model is also trained with 64k context data, but we show that it is able to extrapolate to a context size of 128k in Section 4.2.

For the ablation studies, we used the LLaMA 7B model. It has the same architecture as the newer Llama 2 models except for a shorter pretrained context window size[4], which reduces compute requirements and allows for faster training and evaluations. The training procedure is similar to the 128k models, but we chunk the PG19 dataset into 32k segments instead, and train using $s = 16$ for 400 steps. As shown in Figure 2, YaRN converges faster compared to other interpolation techniques during training and consistently has lower loss.

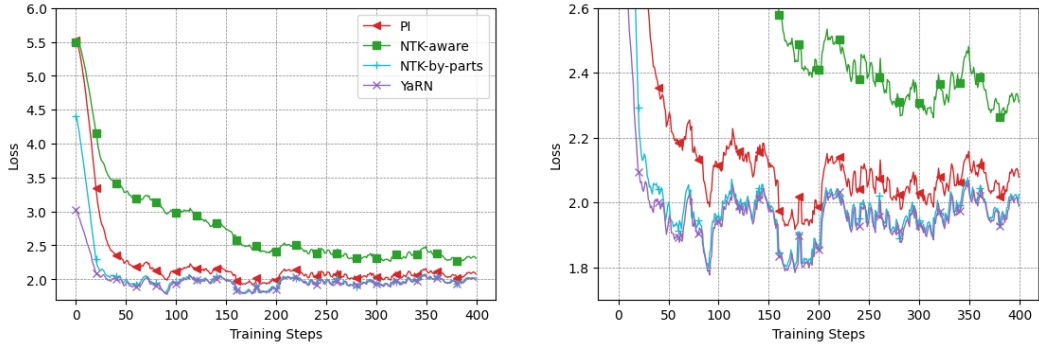

Figure 2: Training loss curves for the LLaMA 7B model extended to 32k context size using different interpolation techniques. The graph on the right is zoomed in.

## 4.2 LONG SEQUENCE LANGUAGE MODELING

To evaluate the long sequence language modeling performances, we use the GovReport (Huang et al., 2021) and Proof-pile (Azerbayev et al., 2022) datasets both of which contain many long sequence samples. For all evaluations, the test splits of both datasets were used exclusively. All perplexity evaluations were calculated using the sliding window method from Press et al. (2022) with $S = 256$, which takes in account the entire documents' perplexity contribution, even if the context window of the model is shorter.

First, we select 10 random samples from Proof-pile with at least 128k tokens each and evaluate the perplexity of each of these samples when truncated at 2k steps from a sequence length of 2k tokens through 128k tokens. Table 1 shows the long sequence performance of fine-tuned Llama 2 $s = 16$ and $s = 32$ models. We demonstrate that YaRN is able to generalize and extrapolate to unseen context lengths and benefit from transfer learning, since the $s = 32$ model was only further trained for 200 steps using the $s = 16$ checkpoint with 64k data and is able to extrapolate to 128k context.

In order to further confirm the effectiveness of YaRN, we compare all four interpolation methods in Figure 3 on the left and Table 5 from Appendix B.1 as an ablation study. YaRN consistently outperforms (has lower perplexity than) other methods in both non fine-tuned and fine-tuned scenarios when using the same number of training steps. We also demonstrate that YaRN has better training efficiency compared to PI in Appendix B.2. More comparisons against open models can be found in Appendix B.3.

## 4.3 PASSKEY RETRIEVAL

The passkey retrieval task as defined in Mohtashami and Jaggi (2023) measures a model's ability to retrieve a simple passkey (i.e., a five-digit number) from amongst a large amount of otherwise meaningless text. For our evaluation of the fine-tuned 32k LLaMA 7B models, we performed

---

[4]LLaMA models have a pretrained context size of 2k tokens, while Llama 2 models have 4k.

| Model Size | Extension Method | Fine-tuned | Training Steps | Extension Scale $s$ | Evaluation Context Window Size | | | | |
|---|---|---|---|---|---|---|---|---|---|
| | | | | | 8192 | 16384 | 32768 | 65536 | 131072 |
| 7B | YaRN | ✓ | 400 | 4k × 16 | 3.51 | 2.99 | 2.65 | 2.42 | $> 10^1$ |
| 7B | YaRN | ✓ | 400+200 | 4k × 32 | 3.56 | 3.04 | 2.70 | 2.45 | 2.37 |
| 13B | YaRN | ✓ | 400 | 4k × 16 | **3.25** | **2.79** | **2.50** | **2.29** | $> 10^1$ |
| 13B | YaRN | ✓ | 400+200 | 4k × 32 | 3.29 | 2.83 | 2.53 | 2.31 | **2.24** |

Table 1: Sliding window perplexity ($S = 256$) of ten 128k Proof-pile documents over Llama 2 models extended via YaRN. We show successful context size extrapolation and transfer learning from 64k to 128k given only 64k context as training data.

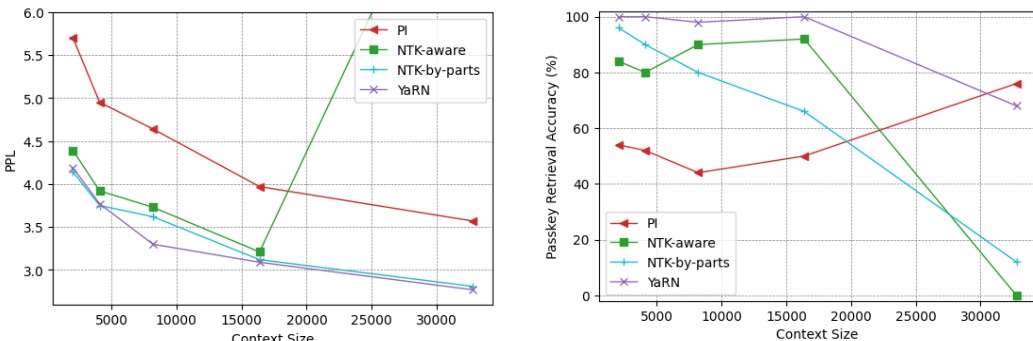

Figure 3: Sliding window perplexity ($S = 256$) of ten 128k Proof-pile documents and passkey retrieval accuracy at different prompt lengths for finetuned LLaMA 7B models fine-tuned to 32k context for 400 steps using different interpolation techniques. YaRN outperforms other interpolation methods given the same training budget.

50 iterations of the passkey retrieval task with the passkey placed at a random location uniformly distributed across the evaluation context window on different prompt lengths ranging from 2k to 32k. YaRN achieves higher scores compared to other interpolation methods when given similar training budget, as seen in Figure 3 on the right. More results and comparisons for Llama 2 models are shown in Appendix B.5.

## 4.4 STANDARDIZED BENCHMARKS

The Hugging Face Open LLM Leaderboard (Hugging Face, 2023) compares a multitude of LLMs across a standardized set of four public benchmarks. Specifically, we use 25-shot ARC-Challenge (Clark et al., 2018), 10-shot HellaSwag (Zellers et al., 2019), 5-shot MMLU (Hendrycks et al., 2021), and 0-shot TruthfulQA (Lin et al., 2022).

To test the degradation of models' short context performance under context extension, we evaluated our Llama 2 and 32k LLaMA 7B models using this suite and compared it to established scores for the baselines. The results are summarized in Table 10 and Table 3. More results for Llama 2 models are shown in Appendix B.6.

We observe that there is minimal performance degradation between the YaRN models and their respective Llama 2 baselines. Some variance is to be expected as the PG19 dataset (Rae et al., 2020) we used for fine-tuning is very different from the original pre-training datased used for LLaMA and Llama 2 models. We also observe that there was on average a 0.49% drop in scores between the YaRN $s = 16$ and $s = 32$ models and can conclude that the the iterative extension from 64k to 128k results in negligible performance loss.

| Extension Method | Fine-tuned | Extension Scale $s$ | ARC-c | Hellaswag | MMLU | TruthfulQA |
|---|---|---|---|---|---|---|
| None | ✗ | - | **51.0** | **77.8** | **35.7** | 34.3 |
| PI | ✓ | 2k × 16 | 44.8 | 70.2 | 25.9 | 34.1 |
| NTK-aware | ✓ | 2k × 16 | 47.4 | 73.9 | 27.7 | 32.6 |
| NTK-by-parts | ✓ | 2k × 16 | 48.5 | 76.6 | 32.7 | 33.4 |
| YaRN | ✓ | 2k × 16 | 48.1 | 77.2 | 30.0 | **35.1** |

Table 2: Performance of context window extensions methods, fine-tuned for 400 steps, on the Hugging Face Open LLM benchmark suite compared with original LLaMA 7B baselines.

| Model Size | Extension Method | Fine-tuned | Extension Scale $s$ | ARC-c | Hellaswag | MMLU | TruthfulQA |
|---|---|---|---|---|---|---|---|
| 7B | None | ✗ | - | 53.1 | 77.8 | 43.8 | **39.0** |
| 7B | YaRN | ✓ | 4k × 16 | 52.3 | 78.8 | 42.5 | 38.2 |
| 7B | YaRN | ✓ | 4k × 32 | 52.1 | 78.4 | 41.7 | 37.3 |
| 13B | None | ✗ | - | **59.4** | 82.1 | **55.8** | 37.4 |
| 13B | YaRN | ✓ | 4k × 16 | 58.1 | **82.3** | 52.8 | 37.8 |
| 13B | YaRN | ✓ | 4k × 32 | 58.0 | 82.2 | 51.9 | 37.3 |

Table 3: Performance of YaRN on the Hugging Face Open LLM benchmark suite compared with original Llama 2 baselines.

## 4.5 COMPUTATIONAL EFFICIENCY

Given that rotary position embeddings are cached during training and inference when the context window size is fixed to a preset length $L$, modifying the interpolation on rotary position embeddings incurs no additional computational or memory cost compared to previous context extension methods, which is the case for all four interpolation methods outlined in this work. YaRN converges the fastest during training compared to other methods, thus is the most computationally efficient, as shown in Table 4.

| Model Size | Model Name | Extension Method | Extension Scale $s$ | Effective Context | Training Time in GPU-Hours (A100) |
|---|---|---|---|---|---|
| 7B | LLaMA YaRN | YaRN | 2k × 16 | 32k | 128 |
| 7B | Llama 2 YaRN | YaRN | 4k × 16 | 64k | 256 |
| 7B | Llama 2 YaRN | YaRN | 4k × 32 | 128k | 256 + 128 |
| 7B | (Chen et al., 2023) | PI | 2k × 8 | 16k | 640 |
| 7B | (Together.ai, 2023) | PI | 4k × 8 | 32k | ? |
| 7B | (Xiong et al., 2023) | NTK-aware | 4k × 44.2 | ≈ 50k | 64000 |
| 7B | (Rozière et al., 2023) | NTK-aware | 4k × 88.6 | ≈ 100k | 6400 |

Table 4: Comparison of training time in A100-hours for different open and closed models using different extension methods.

## 5 CONCLUSION

In conclusion, we have shown that YaRN improves upon all existing RoPE interpolation methods and can act as a drop-in replacement to PI, with no downsides and minimal implementation effort. The fine-tuned models preserve their original abilities on multiple benchmarks while being able to attend to a very large context size. Furthermore, YaRN allows efficient extrapolation with fine-tuning on shorter datasets and can take advantage of transfer learning for faster convergence, both of which are crucial under compute-constrained scenarios. Finally, we have shown the effectiveness of extrapolation with YaRN where it is able to "train short, and test long".

## 6 REPRODUCIBILITY

To aid in reproducibility, we provide, as supplementary material, the entirety of of the code used to train the YaRN models in Table 7, as well as the evaluation code that produced Figure 7 and Tables 6, 7, 10, 8, and 9. The code also contains implementations of various extension methods referenced throughout the paper. For training YaRN, we used the publicly available PG19 dataset (Rae et al., 2020) tokenized to contiguous chunks of 64k tokens.

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

## A    ADDITIONAL DETAILS ON INTERPOLATION METHODS

### A.1    POSITION INTERPOLATION

As mentioned in Section 2.2, PI is one of the earlier works extending context lengths of RoPE. We include some extra details here:

While a direct extrapolation does not perform well on sequences $w_1, \cdots, w_L$ with $L$ larger than the pre-trained limit, they discovered that interpolating the position indicies within the pre-trained limit works well with the help of a small amount of fine-tuning. Specifically, given a pre-trained language model with RoPE, they modify the RoPE by

$$f'_{\boldsymbol{W}}\left(\boldsymbol{x}_m, m, \boldsymbol{\theta}\right) = f_{\boldsymbol{W}}\left(\boldsymbol{x}_m, \frac{mL}{L'}, \boldsymbol{\theta}\right), \tag{16}$$

where $L' > L$ is a new context window beyond the pre-trained limit. With the original pre-trained model plus the modified RoPE formula, they fine-tuned the language model further on several orders of magnitude fewer tokens (a few billion in Chen et al. (2023)) and successfully acheived context window extension.

### A.2    DETAILS OF "NTK-AWARE" INTERPOLATION

In Section 3.1, we introduce a change of basis from $b$ to $b'$ in the definition of "NTK-aware" interpolation method.

Precisely, following the notations set out in Section 2.1 Eq. 7, we define the "NTK-aware" interpolation scheme as follows:

**Definition 3** *The "NTK-aware" interpolation is a modification of RoPE using Eq. 7 with the following functions, given $s$ as the scale factor.*

$$g(m) = m \tag{17}$$

$$h(\theta_d) = {b'}^{-2d/|D|}, \tag{18}$$

*where*

$$b' = b \cdot s^{\frac{|D|}{|D|-2}} \tag{19}$$

$$s = \frac{L'}{L}. \tag{20}$$

Given the results from (bloc97, 2023a), this method performs much better at extending the context size of non-fine-tuned models compared to PI (Chen et al., 2023). However, one major disadvantage of this method is that given it is not just an interpolation scheme, some dimensions are slightly extrapolated to "out-of-bound" values, thus fine-tuning with "NTK-aware" interpolation (bloc97, 2023a) yields inferior results to PI (Chen et al., 2023). Furthermore, due to the "out-of-bound" values, the theoretical scale factor $s$ does not accurately describe the true context extension scale. In practice, the scale value $s$ has to be set higher than the expected scale for a given context length extension.

The mathematical derivation of the base change is the following:

Recall that our goal is to spread out the interpolation pressure across the hidden dimensions using a base-change instead of scaling the frequencies by a fixed factor $s$. The property we want to guarantee is that: The lowest frequency needs to be scaled as much as linear positional scaling and the highest frequency to stay constant.

We introduce a new base $b'$ such that the last dimension matches the wavelength of linear interpolation with a scale factor $s$. Since the original RoPE method skips odd dimensions in order to concatenate both $\cos(\frac{2\pi x}{\lambda})$ and $\sin(\frac{2\pi x}{\lambda})$ components into a single embedding, the last dimension $d \in D$ is $|D| - 2$.

The new base $b'$ can be chosen so that

$${b'}^{\frac{|D|-2}{|D|}} = s \cdot b^{\frac{|D|-2}{|D|}}. \tag{21}$$

Solving for $b'$ yields

$$b' = b \cdot s^{\frac{|D|}{|D|-2}}. \tag{22}$$

## A.3  THE IMPACT OF PRE-SOFTMAX SCALING OF YARN ON PERPLEXITY

In Section 3.3, we mention the impact of the factor $t$ inside the softmax computation of attention weights. Here we fix 896 16k-token documents from RedPajama (Computer, 2023)[5], and calculate their perplexity scores with different scaling $1/\sqrt{t}$. The result is in Figure 4. For comparison, recall that our recommended factor in this case ($s = 8$) is given by the following.

$$\sqrt{\frac{1}{t}} = 0.1\ln(s) + 1 \approx 1.208. \tag{23}$$

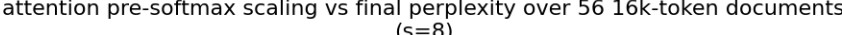

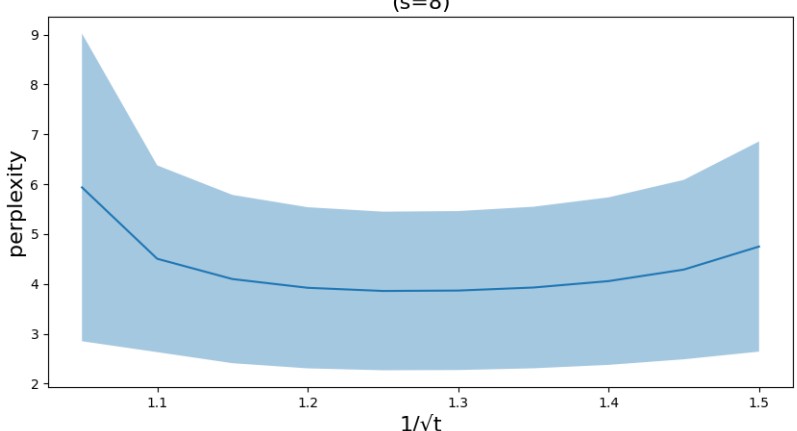

Figure 4: Fix $s = 8$, compare the LLaMA 7b perplexity on 896 16k-token documents over different scaling $1/\sqrt{t}$. The shaded area represents 1 standard deviation (68%).

To show the impact of the factor $1/\sqrt{t}$ on different token positions, we cut each 16k-token document into chunks of 2048 tokens, and further plot the mean perplexity change comparing to $t = 1$ in percentages

$$\frac{\text{ppl}(t) - \text{ppl}(t = 1)}{\text{ppl}(t = 1)} \tag{24}$$

of each chunk. The plot is shown in Figure 5.

---

[5]We choose RedPajama because it is the open-source dataset closest to the training dataset of LLaMA as far as we are aware of.

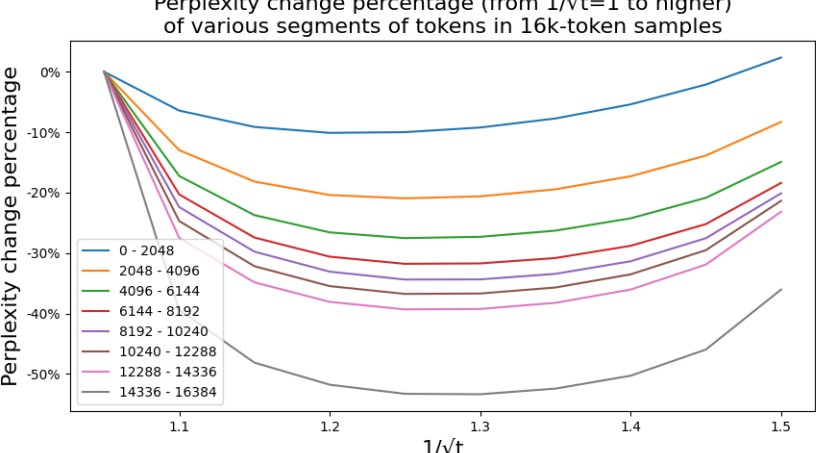

Figure 5: Fix $s = 8$, compare the mean of perplexity change percentages $\dfrac{\mathrm{ppl}(t) - \mathrm{ppl}(t = 1)}{\mathrm{ppl}(t = 1)}$ at different segments of token positions on 896 16k-token documents over different scaling $1/\sqrt{t}$.

To further demonstrate the best values of $t$ across all samples over different token positions, we plot the sample counts with minimal perplexity at a given $1/\sqrt{t}$ for each of the 8 position segments over the 16k-token range in Figure 6.

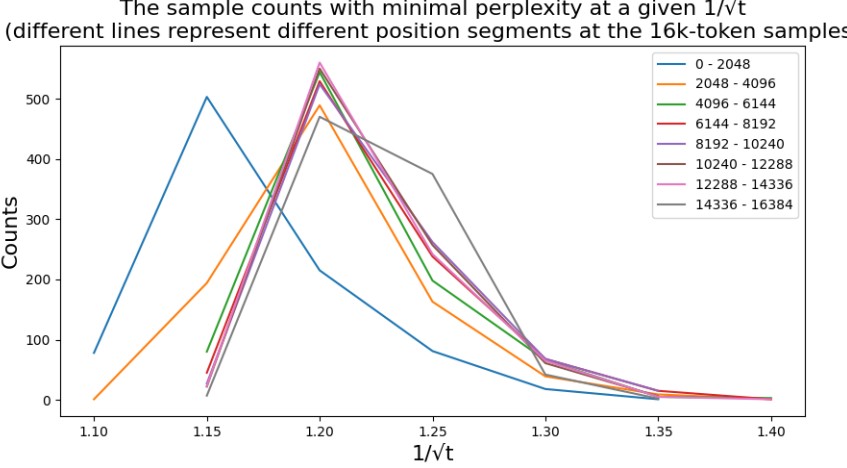

Figure 6: The sample counts (out of the 896 samples) with minimal perplexity at a given $1/\sqrt{t}$ for a given segment of token positions over the 16k-token range.

We observe that:

- for a suitable $t$, a sample may obtain better perplexity scores across the extended context window;

- the best value of $t$ is mostly consistent across different samples and different positions.

We remark that this finding is consistent for different values of $s$ and the best value of $t$ follows our recommended formula (Eq. 15) closely.

# B ADDITIONAL TABLES AND CHARTS

## B.1 ABLATION STUDY

| Extension Method | Fine-tuned | Training Steps | Extension Scale $s$ | Evaluation Context Window Size | | | | |
| --- | --- | --- | --- | --- | --- | --- | --- | --- |
| | | | | 2048 | 4096 | 8192 | 16384 | 32768 |
| None | ✗ | - | - | **4.05** | - | - | - | - |
| PI | ✗ | - | 2k × 2 | 4.36 | 3.90 | - | - | - |
| NTK-aware | ✗ | - | 2k × 2 | 4.08 | 5.97 | - | - | - |
| NTK-by-parts | ✗ | - | 2k × 2 | 4.12 | 3.71 | | - | - |
| YaRN | ✗ | - | 2k × 2 | 4.07 | **3.67** | - | - | - |
| PI | ✗ | - | 2k × 4 | 7.09 | 6.39 | 6.18 | - | - |
| NTK-aware | ✗ | - | 2k × 4 | 4.27 | 3.84 | $> 10^1$ | - | - |
| NTK-by-parts | ✗ | - | 2k × 4 | 4.39 | 4.03 | 4.11 | - | - |
| YaRN | ✗ | - | 2k × 4 | 4.19 | 3.77 | 3.65 | - | - |
| PI | ✗ | - | 2k × 8 | $> 10^1$ | $> 10^1$ | $> 10^1$ | $> 10^1$ | - |
| NTK-aware | ✗ | - | 2k × 8 | 4.64 | 4.27 | 4.24 | $> 10^1$ | - |
| NTK-by-parts | ✗ | - | 2k × 8 | 4.98 | 4.91 | 5.33 | 5.79 | - |
| YaRN | ✗ | - | 2k × 8 | 4.37 | 3.95 | 3.81 | 3.33 | - |
| PI | ✗ | - | 2k × 16 | $> 10^2$ | $> 10^2$ | $> 10^2$ | $> 10^2$ | $> 10^2$ |
| NTK-aware | ✗ | - | 2k × 16 | 5.23 | 5.02 | 5.22 | 6.85 | $> 10^1$ |
| NTK-by-parts | ✗ | - | 2k × 16 | 6.04 | 7.54 | $> 10^1$ | $> 10^1$ | $> 10^1$ |
| YaRN | ✗ | - | 2k × 16 | 4.61 | 4.24 | 4.18 | 3.66 | 3.45 |
| PI | ✗ | - | Dynamic | **4.05** | 3.90 | 6.18 | $> 10^1$ | $> 10^2$ |
| NTK-aware | ✗ | - | Dynamic | **4.05** | 5.97 | $> 10^1$ | $> 10^1$ | $> 10^1$ |
| NTK-by-parts | ✗ | - | Dynamic | **4.05** | 3.71 | 4.11 | 5.79 | $> 10^1$ |
| YaRN | ✗ | - | Dynamic | **4.05** | **3.67** | 3.65 | 3.33 | 3.45 |
| PI | ✓ | 400 | 2k × 16 | 5.70 | 4.95 | 4.64 | 3.97 | 3.57 |
| NTK-aware | ✓ | 400 | 2k × 16 | 4.39 | 3.92 | 3.73 | 3.21 | 8.49 |
| NTK-by-parts | ✓ | 400 | 2k × 16 | 4.14 | 3.75 | 3.62 | 3.12 | 2.81 |
| YaRN | ✓ | 400 | 2k × 16 | 4.19 | 3.77 | **3.30** | **3.09** | **2.77** |

Table 5: Sliding window perplexity ($S = 256$) of ten 128k Proof-pile documents over the LLaMA 7B model extended via different methods.

## B.2 TRAINING EFFICIENCY OF YARN

Table 6 shows a side-by-side comparison of the Llama 2 7B model extended from $4096$ to $8192$ context length via PI (LLongMA-2 7B[6]) and YaRN. Note that the PI model was trained using the methodology in Chen et al. (2023), while YaRN used the same methodology but 2.5x less training steps and data, as described in Section 4. Even if YaRN was only fine-tuned for 400 steps compared to PI's 1000 steps, we obtain similar results to PI.

| Model Size | Extension Method | Fine-tuned | Training Steps | Extension Scale $s$ | Evaluation Context Window Size | | | |
|---|---|---|---|---|---|---|---|---|
| | | | | | 2048 | 4096 | 6144 | 8192 |
| 7B | None | ✗ | - | - | 4.00 | 3.58 | - | - |
| 7B | PI | ✗ | - | 4k $\times$ 2 | 4.30 | 3.84 | 3.83 | 3.65 |
| 7B | YaRN | ✗ | - | 4k $\times$ 2 | 4.03 | 3.61 | _3.60_ | 3.49 |
| 7B | PI | ✓ | 1000 | 4k $\times$ 2 | _3.92_ | _3.51_ | **3.51** | **3.34** |
| 7B | YaRN | ✓ | 400 | 4k $\times$ 2 | **3.91** | **3.50** | **3.51** | _3.35_ |

Table 6: Sliding window perplexity ($S = 256$) of ten 128k Proof-pile documents over the Llama-2 7b model extended via PI and YaRN with different training steps. YaRN obtains comparable results to PI using much less training steps.

## B.3 COMPARING THE PERPLEXITY OF VARIOUS METHODS OVER A SLIDING WINDOW

We further evaluated the Llama 2 models fine-tuned using YaRN at the scale factor $s = 16, 32$ and compared them against a few long-context open-source models fine-tuned from Llama-2 such as Together.ai (Together.ai, 2023) and "NTK-aware" Code Llama (Rozière et al., 2023). The results are summarized in Table 7 (with a more detailed plot in Figure 7).

| Model Size | Model Name | Context Window | Extension Method | Evaluation Context Window Size | | | | |
|---|---|---|---|---|---|---|---|---|
| | | | | 8192 | 32768 | 65536 | 98304 | 131072 |
| 7B | Together | 32k | PI | **3.50** | **2.64** | $> 10^2$ | $> 10^3$ | $> 10^4$ |
| 7B | Code Llama | 100k | NTK | 3.71 | 2.74 | 2.55 | 2.54 | 2.71 |
| 7B | YaRN ($s = 16$) | 64k | YaRN | 3.51 | 2.65 | **2.42** | $> 10^1$ | $> 10^1$ |
| 7B | YaRN ($s = 32$) | 128k | YaRN | 3.56 | 2.70 | 2.45 | **2.36** | **2.37** |
| 13B | Code Llama | 100k | NTK | 3.54 | 2.63 | 2.41 | 2.37 | 2.54 |
| 13B | YaRN ($s = 16$) | 64k | YaRN | **3.25** | **2.50** | **2.29** | $> 10^1$ | $> 10^1$ |
| 13B | YaRN ($s = 32$) | 128k | YaRN | 3.29 | 2.53 | 2.31 | **2.23** | **2.24** |

Table 7: Sliding window perplexity ($S = 256$) of ten 128k Proof-pile documents truncated to evaluation context window size

We observe that the model exhibits strong performance across the entire targeted context size, with YaRN interpolation being the first method to successfully extend the effective context size of Llama 2 to 128k.

Furthermore, in Appendix B.4, we show the results of the average perplexity on 50 untruncated GovReport documents with at least 16k tokens per sample evaluated on the setting of 32k maximal context window without Dynamic Scaling in Table 8. Similar to the Proof-pile results, the GovReport results show that fine-tuning with YaRN achieves good performance on long sequences.

Table 7 summarizes the results and a visualized and more detailed view is presented in Figure 7 here.

---

[6]LLongMA-2 7B (Quesnelle et al., 2023) is fine-tuned from Llama 2 7B, trained at 8k context length with PI using the RedPajama dataset (Computer, 2023).

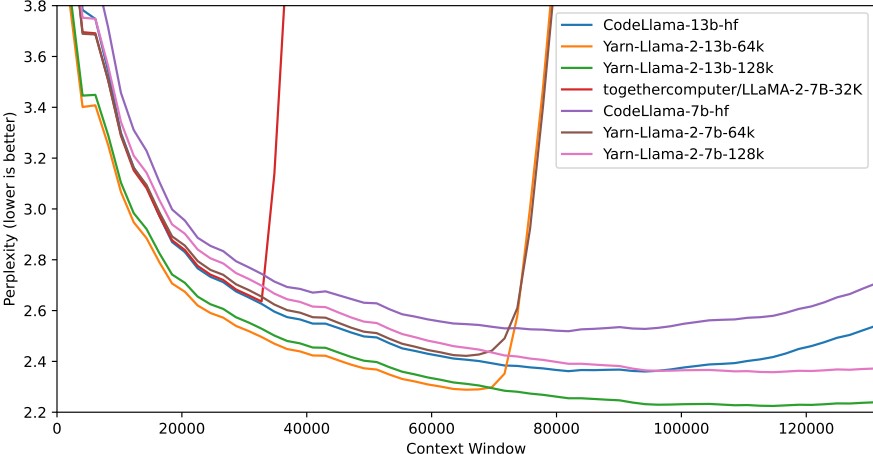

Figure 7: Sliding window perplexity ($S = 256$) of a 1.28M-token Proof-pile documents truncated to the context window size of the fine-tuned model

## B.4 GOVREPORT EVALUATIONS

In Section 4.2, we mention the evaluation on GovReport documents. The evaluation results are detailed in Table 8 below.

| Model Size | Model Name | Context Window | Extension Method | Perplexity |
|---|---|---|---|---|
| 7B | Together | 32k | PI | 3.67 |
| 7B | Code Llama | 100k | NTK | 4.44 |
| 7B | YaRN ($s = 16$) | 64k | YaRN | **3.59** |
| 7B | YaRN ($s = 32$) | 128k | YaRN | 3.64 |
| 13B | Code Llama | 100k | NTK | 4.22 |
| 13B | YaRN ($s = 16$) | 64k | YaRN | **3.35** |
| 13B | YaRN ($s = 32$) | 128k | YaRN | 3.39 |

Table 8: Sliding window perplexity ($S = 256$) of 50 long GovReport documents with a fixed context window size of 32k

## B.5 PASSKEY RETRIEVAL

For our evaluation of the 64k and 128k models, we performed 10 iterations of the passkey retrieval task with the passkey placed at a random location uniformly distributed across the evaluation context window on different context window sizes ranging from 8k to 128k. Both 7b and 13b models fine-tuned using YaRN at 128k context size passes the passkey retrieval task with very high accuracy ($> 99\%$) within the entire context window size.

| Model Size | Model Name | Scaling Factor ($s$) | Context Window | Training Data Context | Extension Method | Passkey Context | Passkey Accuracy |
|---|---|---|---|---|---|---|---|
| 7B | Together | 4 | 32k | 32k | PI | 32k | 100% |
| 7B | Code Llama | 88.6 | 100k | 16k | NTK | 112k | 94.3% |
| 7B | YaRN | 16 | 64k | 64k | YaRN | 64k | 96.3% |
| 7B | YaRN | 32 | 128k | 64k | YaRN | 128k | 99.4% |
| 13B | Code Llama | 88.6 | 100k | 16k | NTK | 128k | 99.4% |
| 13B | YaRN | 16 | 64k | 64k | YaRN | 64k | 97.5% |
| 13B | YaRN | 32 | 128k | 64k | YaRN | 128k | 99.4% |

Table 9: Passkey retrieval performance of various models. The passkey context denotes the maximum tested context window size where the accuracy of passkey retrieval was $>= 80\%$, and the passkey accuracy is the average accuracy of passkey retrieval on all context sizes tested that were smaller or equal than the passkey context size.

Here we can observe that the lowest perplexity point alone does not provide a comprehensive depiction on the "effective context size" that an LLM can attend to. While the Code Llama 13b model exhibits increasing perplexity above 100k context lengths, it was still able to accurately retrieve the passkey at a context length of 128k. This suggest that while the output of Code Llama might start to degrade in quality above 100k context size, it is still able to maintain strong retrieval capabilities.

In addition, as YaRN with $s = 32$ was trained for 200 more steps than YaRN with $s = 16$ while having a higher passkey accuracy with similar perplexity, we hypothesize that perplexity may not be a great indicator of whether an LLM is able to attend to all tokens and does not exhaustively determine long context performance. This also suggests that the YaRN models with $s = 16$ might be relatively undertrained for the passkey retrieval task.

### B.6 STANDARDIZED BENCHMARKS

To test the degradation of model performance under context extension, we evaluated our models using this suite and compared it to established scores for the Llama 2 baselines as well as publicly available PI and "NTK-aware" models.

| Model Size | Model Name | Context Window | Extension Method | ARC-c | Hellaswag | MMLU | TruthfulQA |
|---|---|---|---|---|---|---|---|
| 7B | Llama 2 | 4k | None | **53.1** | 77.8 | **43.8** | 39.0 |
| 7B | Together | 32k | PI | 47.6 | 76.1 | 43.3 | **39.2** |
| 7B | Code Llama | 100k | NTK-a | 39.9 | 60.8 | 31.1 | 37.8 |
| 7B | YaRN ($s = 16$) | 64k | YaRN | 52.3 | **78.8** | 42.5 | 38.2 |
| 7B | YaRN ($s = 32$) | 128k | YaRN | 52.1 | 78.4 | 41.7 | 37.3 |
| 13B | Llama 2 | 4k | None | **59.4** | 82.1 | **55.8** | 37.4 |
| 13B | Code Llama | 100k | NTK-a | 40.9 | 63.4 | 32.8 | **43.8** |
| 13B | YaRN ($s = 16$) | 64k | YaRN | 58.1 | **82.3** | 52.8 | 37.8 |
| 13B | YaRN ($s = 32$) | 128k | YaRN | 58.0 | 82.2 | 51.9 | 37.3 |

Table 10: Performance of context window extensions methods on the Hugging Face Open LLM benchmark suite compared with original Llama 2 baselines

### B.7 DYNAMIC SCALING ON MODELS WITHOUT ANY FINE-TUNING

We first recall from Section 3.4 that the Dynamic Scaling technique is an inference-time technique that dynamically update the factor $s$ in interpolation methods such as PI, "NTK-by-parts" and YaRN. We choose the original Llama 2, fix a sample in GovReport and calculate its perplexity on a sliding window of 256 tokens using RoPE, Dynamic-PI and Dynamic-YaRN.

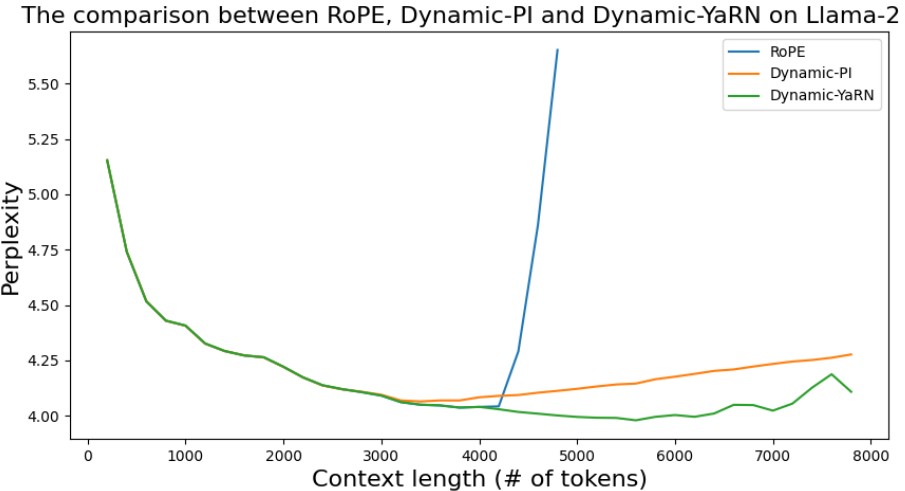

Figure 8: The comparison between RoPE, Dynamic-PI and Dynamic-YaRN using Llama 2 on a long GovReport sample. This model has not been finetuned for long context.

Since the original maximal context length of Llama 2 is $4096$, we observe that Dynamic Scaling effectively extend the inference length and Dynamic-YaRN achieves better performance than Dynamic-PI. The resulting chart is in Figure 8.

We see that

- Dynamic Scaling effectively prevents the blow-up of perplexity score beyond pretrained context window;
- Dynamic-YaRN outperforms Dynamic-PI in terms of long-range perplexity on pretrained Llama-2 without any finetuning.

