# OpenReview forum: "YaRN: Efficient Context Window Extension of Large Language Models"
_ICLR.cc/2024/Conference — ICLR 2024 poster_

### Official Review · Reviewer_swin · 2023-10-30

**Soundness:** 3 good
**Presentation:** 2 fair
**Contribution:** 3 good
**Rating:** 6
**Confidence:** 2

**Summary:**

This paper proposed an encoding scheme that could change the after training context window size of a model without sacrificing performance.

**Strengths:**

1. Motivated and written well.
2. An important problem to work on.

**Weaknesses:**

1. Connection to NTK stuff is very difficult to understand and somehow not convincing. I feel the only thing authors want to say is we shouldn't treat all dimensions equally, which IMO is already a good idea. Connecting to NTK stuff actually confused me quite a bit.

2. Part of baseline in experiments could be missing.

**Questions:**

In page 4 you mentioned RoPE can't retain high frequency information, but you also say RoPE closely resembles Fourier Features (Tancik et al., 2020) in many aspects, as it is possible to define RoPE as a special 1D case of a Fourier Feature. But somehow I don't know why Fourier Feature can't learn high frequency stuff. Fourier Feature is just trying to map the basis to all sort of features and your description here is fairly confusing. Without a full proper explanation, I think the motivation will be fairly questionable then.

2. Is there a training-free context window extension stuff that should be included?
    At least the baseline should be just using encoding scheme as in T5 that we could just change the context window size. For certain task I tried, it doesn't really affect the performance much. So this should be treated as a baseline as well.

3. Can you also add the real "clock-time" of training of these 3 methods (PI, NTK, YaRN) so we will have a better understanding what's the cost of using it?

---

> ### Author Response · Authors · 2023-11-21
>
> Thank you very much for your valuable insights. We will improve the explanations about NTK-aware and NTK-by-parts in order to connect them better to YaRN. The original justification for NTK-aware scaling was definitively taken from the lessons learned from the Fourier Features paper and NTK theory. Without that paper, we would not have thought of the initial seed idea on YaRN. Also, we will be providing a much better baseline in the next draft, where we will compare all four methods after and before fine-tuning with the same hyperparameters and data.
>
> ## Regarding the questions:
>
> We will improve the explanations on the connections of NTK-aware/by-parts interpolation to NTK theory and Fourier Features.
>
> The quick explanation on its motivation is that given that RoPE can be expressed similarly to Fourier Features, and we know that linearly interpolating the RoPE up to a certain scale would significantly degrade the model’s performance, even after fine-tuning. The reason behind it is explained much better in the cited Fourier Features paper, but in short if they showed that high-frequency components in Fourier Features are critical for networks to learn the embeddings properly, the same should apply to RoPE.
> This is also somewhat corroborated by a lack of extremely large context sized models that use PI. We are pretty sure that most ML labs out there would have tried and failed to train a PI model of >=64k context size as the degradation from it is unrecoverable. All open models that have bigger than 32k context either use NTK-aware or YaRN interpolation. We initially only tried to train a 64k and 128k model using YaRN and not PI because we knew that PI would not have worked at such extreme scalings.
>
> YaRN with Dynamic Scaling works somewhat well for up to s=4 (that is 4x the original context size) without any fine-tuning. We will add more PPL tests and ablations that include Dynamic Scaling in the next version.
>
> Finally, the clock-time of training these three methods are identical if we do not care about the final results. The three interpolation methods do not add any additional compute or memory overhead during training, and they are also free during inference. The only difference is that YaRN converges much faster (and also achieves a lower final loss), and we will provide training convergence comparisons in the next version. As a quick comparison though, the context extension part of Code Llama was fine-tuned using roughly 6400 A100-hours of time, while YaRN 128k model only used 380 A100-hours of compute. We know that this comparison is not optimal as Code Llama was fine-tuned on code, so we will provide a better baseline using PI.

---

### Official Review · Reviewer_HiJG · 2023-11-01

**Soundness:** 3 good
**Presentation:** 3 good
**Contribution:** 3 good
**Rating:** 8
**Confidence:** 4

**Summary:**

The paper presents YaRN (Yet another RoPE extensioN method), a compute-efficient method to extend the context window of large language models (LLMs) trained with Rotary Position Embeddings (RoPE). YaRN requires 10x less tokens and 2.5x less training steps than previous methods. The method combines attention scaling, "NTK-by-parts" interpolation, and dynamic scaling to achieve state-of-the-art performance in context window extensions. YaRN is compatible with libraries that modify the attention mechanism, such as Flash Attention, and allows for efficient extrapolation with fine-tuning on shorter datasets.

**Strengths:**

- YaRN achieves state-of-the-art performance in context window extensions with significantly fewer training steps and tokens compared to previous methods.
- The method is compatible with libraries that modify the attention mechanism, such as Flash Attention 2, making it more versatile for various use cases. It provides a drop-in replacement for Position Interpolation (PI) with no downsides and minimal implementation effort.
- YaRN allows for efficient extrapolation with fine-tuning on shorter datasets, enabling faster convergence in compute-constrained scenarios. The paper also demonstrates the capability of YaRN to extrapolate beyond the limited context of a fine-tuning dataset, showcasing its ability to generalize to unseen context lengths.
- The experimental design of the paper is scientific. The authors evaluate YaRN on various benchmarks, including long sequence language modeling, passkey retrieval, and standardized benchmarks. The results consistently show that YaRN outperforms previous methods in context window extensions, supporting the author's claims.

**Weaknesses:**

- The paper does not provide a comprehensive comparison of YaRN with other length extending methods in terms of computational efficiency and memory usage.
- In the Related Work Section, the authors mention two works, ReRoPE and LM-Infinite, which tackle the same target in terms of extending LLM sequence length, and claim that they can extend to infinite length without severe loss expansion. If that is true, the contribution of YaRN will be greatly reduced (i.e., extend to 16k). Also, I am interested in the results of these two works compared to the proposed YaRN. I know due to the attention modification, they are not supported directly by Flash Attention. However, they represent a series of length extending methods. It would be better to compare them as well even under a small but fair setting.
- Please provide more detailed data points or loss curves between PI and YaRN along with training tokens. Is it possible that with the same tokens, PI will converge faster than YaRN?

**Questions:**

- In Table 5, for the PASSKEY retrieval experiment, why not keep all the settings the same? Now, 7B-PI is trained on 32k and 7B-YaRN is trained on 64k, it is difficult to compare and draw reasonable conclusions. Also, how do you explain that the PI method can get 100% ACC in terms of key retrieval, while YaRN can only get 96.3% ACC (see row 1 and 3 in Table 5), does this mean that YaRN may lose some ability compared to the PI method? Another question is, is the total number of tokens trained in all experiments in Table 5 the same?
- In what aspects or tasks does the PI method have advantages over YaRN?
- In what aspects or tasks does the NTK method have advantages over YaRN?
- Under what circumstances should we use PI? Under what circumstances should we modify the base of rope?

---

> ### Author Response · Authors · 2023-11-21
>
> Thank you so much for the detailed review and the encouraging rating! We would like to first point out that we have significantly revised the paper and added in many new ablations, charts and tables. We are wrapping up with the new version but it is still not fully ready at the time of this comment.
> We will include a new chart:
> - Training loss curves for the LLaMA 7B model extended to 32k context size using different interpolation techniques.
> and a few new tables:
> - Sliding window perplexity (S = 256) of ten 128k Proof-pile documents over the LLaMA 7b model across PI, NTK-aware, NTK-by-parts, YaRN with various context length, fine-tuned and non-fine-tuned.
> - Performance of PI, NTK-aware, NTK-by-parts, YaRN on the Hugging Face Open LLM benchmark suite compared with original LLaMA 7B baselines.
> - Comparison of training time in A100-hours for different models and extension methods.
>
> ## Regarding the weakness:
>
> - This is a good point. We will add a table comparing A100-hours to hopefully address the comparison of compute efficiency between different methods.
> - Given the significant compute needs of running long context benchmarks, these alternate methods that do not support Flash Attention are currently low priority for us, as they have no chance to achieve 64k or 128k context sizes with a “reasonable” single-node server machine (eg. 8xA100s). Furthermore, YaRN is unique in a sense that the higher the context size extension is, the better it performs compared to other methods. It would not be appropriate to compare a fine-tuned YaRN model to non fine-tuned models at lower context sizes as the differences will either be minimal compared to other methods that support fine-tuning or simply be unfair to methods that do not support efficient fine-tuning. The results would either be inconclusive (difference is too insignificant) or YaRN would destroy them as we enjoy the benefit of a very quick fine-tune.
> - We will add a training loss plot including PI, NTK-aware, NTK-by-parts, and YaRN for different training steps.
>
> ## Regarding the questions:
>
> - This is a very good point and we really appreciate this. We are working on better PASSKEY experiments fixing the extended context length across different methods. By re-running the experiments over models fine-tuned over the same context length, we no longer observe such a drop. We suspect that the original drop from PI 100% to YaRN 96.3% is mostly affected by the difference of fine-tuned context length 32k vs 64k.
> - The PI and NTK methods are easier to implement, whereas YaRN seems to outperform both in all situations with better data efficiency.

---

### Official Review · Reviewer_t3XG · 2023-11-02

**Soundness:** 3 good
**Presentation:** 2 fair
**Contribution:** 3 good
**Rating:** 6
**Confidence:** 4

**Summary:**

This paper presents YaRN, a method to efficiently extend the context window of large language models pretrained with Rotary Position Embeddings (RoPE). The key contributions are:

- Identifying issues with existing interpolation methods like Position Interpolation (PI) and proposing solutions:
    - "NTK-aware" interpolation to preserve high frequency information
    - "NTK-by-parts" interpolation to maintain local distance relationships
    - Dynamic scaling for inference-time interpolation
- Introducing an attention temperature hyperparameter for entropy control
- Combining these techniques into YaRN, which extends context 10x more efficiently than prior work

The method is evaluated by fine-tuning LLaMA models and testing on long-context perplexity, retrieving passkeys, and standard benchmarks. Results show YaRN matches or exceeds prior context extension techniques.

**Strengths:**

- The paper clearly identifies limitations of prior work on extending context for RoPE models and proposes principled solutions grounded in theory. This shows strong technical understanding.

- The YaRN method is innovative in how it combines multiple solutions in a synergistic way to efficiently extend context. The temperature parameter for controlling entropy is also novel.

- The experiments comprehensively evaluate perplexity on long sequences, passkey retrieval, and standardized benchmarks. The results demonstrate YaRN's capabilities for few-shot context extension and extrapolation.

- The writing is clear and well-structured. The background builds intuition and the methodology explains each component of YaRN in a logical progression.

**Weaknesses:**

- The ablation study evaluating the impact of each proposed technique in isolation would help validate their necessity in YaRN.

- More analysis and intuition explaining why the temperature parameter works would strengthen that contribution.

**Questions:**

- For the temperature parameter, did you try other hyperparameters or learning it? Why is attention temperature most effective?

- You mention the temperature provides a uniform perplexity improvement over the context window. Can you elaborate why you think this is the case?

- Can you include results quantifying the impact of each proposed technique independently?

---

> ### Author Response · Authors · 2023-11-21
>
> Thank you for the thoughtful comments. We would like to first point out that we have significantly revised the paper and added in many new ablations, charts and tables. We are wrapping up with the new version but it is still not fully ready at the time of this comment.
> We will include a new chart:
> - Training loss curves for the LLaMA 7B model extended to 32k context size using different interpolation techniques.
> and a few new tables:
> - Sliding window perplexity (S = 256) of ten 128k Proof-pile documents over the LLaMA 7b model across PI, NTK-aware, NTK-by-parts, YaRN with various context length, fine-tuned and non-fine-tuned.
> - Performance of PI, NTK-aware, NTK-by-parts, YaRN on the Hugging Face Open LLM benchmark suite compared with original LLaMA 7B baselines.
> - Comparison of training time in A100-hours for different models and extension methods.
>
> ## Regarding the weakness:
>
> - We will include more ablations and improved the rigorousness in terms of control variables. The major new ones are described above.
> - In terms of why the temperature works, we originally did have a theory touching the entropy of attention weights and supported by limited experiments. But as we expand the ablation, we found that the entropy of attention weights in deeper layers behave in a way contradicting our theory. Unfortunately, we are still trying to figure out the theory side.
>
> ## Regarding the questions:
>
> - For other hyperparameters, we are not aware of many other choices inside RoPE. As stated above, why attention temperature is the most effective is still open. The temperature parameters can in theory be learned is a very good point and we have thought about it during our work. Since it amounts to uniform scaling on q and k, we believe the model eventually learns the temperature scaling after sufficient fine-tuning, but hard-coding it has the benefit of fine-tuning on less data (as suggested in Table 1)
> - We do not have a mathematical justification. From the experiments, all we can conclude is that the improvement coming from the temperature scaling seems to be independent of the exact batch of data used during forward pass. If we freeze all other parameters and only consider the temperature scaling, it might suggest that the model is undertrained on this specific variable. A more rigorous explanation remains open and is rather intriguing.
> - We will include a more comprehensive table including PI, NTK-aware, NTK-by-parts and YaRN with different fixed context length, fine-tuned and non-fine-tuned.

---

### Official Review · Reviewer_x5Kh · 2023-11-02

**Soundness:** 2 fair
**Presentation:** 2 fair
**Contribution:** 3 good
**Rating:** 6
**Confidence:** 4

**Summary:**

This paper proposes a way of modifying RoPE embeddings so that they extrapolate better to sequences longer than the model was originally trained on. The YaRN method leverages a recent (unpublished) method, called "NTK-by-parts" [1], as well as modifying the default temperature value of the attention softmax. The NTK-by-parts method is based on the following observation: Performing position interpolation [2] on the high-frequency (small wavelength) RoPE rotation angles could make it such that neighboring tokens have rotation angles that are very difficult for the model to distinguish, thus potentially confusing the model regarding the relative positions of those tokens. Thus, the NTK-by-parts method functions by determining the amount of position interpolation to perform based on the frequency of the RoPE rotation angles: no interpolation is used for high-frequency angles, while standard interpolation is used for the low-frequency angles (and "middle" frequency angles are handled with some interpolation). Once these updates to the RoPE embeddings (and attention temperature) are made, the whole model is fine-tuned on a small amount of data (~0.1% of the pre-training corpus).

The YaRN method is shown to perform relative well relative to baseline RoPE extension methods across various experiments (e.g., long sequence language modeling, and several standard LLM benchmark tasks from HuggingFace Open LLM leaderboard).

- [1] https://github.com/jquesnelle/yarn/pull/1
- [2] S. Chen, S. Wong, L. Chen, and Y. Tian. Extending context window of large language models via
positional interpolation, 2023. arXiv: 2306.15595.

**Strengths:**

- The proposed method appears to perform well relative to a few baseline RoPE extension methods.
- The proposed method is relatively well-motivated --- it seems like a reasonable idea to not interpolate for high-frequency rotation angles.
- The proposed method has already been used in open-source LLMs in industry.

**Weaknesses:**

- I found the paper relatively difficult to follow. I think the method could be presented in a much simpler and more direct manner. The "NTK-aware interpolation" could likely be moved to the appendix, as it is not part of the YaRN method. The background section could be significantly shortened (currently 1.5 pages).
- I think the experimental results could be much more thorough, and much more clearly presented.
  - Including more baselines throughout all the experimental results: Standard RoPE embeddings (e.g., Llama-2), ALiBi, T5 relative position embeddings, absolute position embeddings, Position Interpolation (PI), NTK-aware interpolation, YaRN (with and without fine-tuning, with and without temperature scaling, with static vs. dynamic scaling, different scale factors, etc.).
  - Replacing tables with plots (similar to Figure 4 in appendix, but with clearer+more baselines). The tables are more difficult to interpret. And the plots show results for short sequences as well as long sequences, which is helpful for understanding the performance of the method.
  - Adding detailed ablations for YaRN, as mentioned above (with and without fine-tuning, with and without temperature scaling, with static vs. dynamic scaling, different scale factors, etc.).
  - Choosing better baselines. Why are Together and Code Llama chosen as the main baselines? Why is PI not included in every table?

Overall, given the issues with the presentation of the method and (most importantly) the experimental results, I have currently chosen "marginal reject" (I was torn between "marginal accept" and "marginal reject"). While I think the community could benefit from seeing the proposed idea (in particular, the part about only doing interpolation for low-frequency RoPE angles), I think there are currently too many open questions related to the results, and how this method compares with baselines (and with itself, with ablations), to accept the paper in the current form to a peer-reviewed conference. Open to being swayed though, given the potential of this line of work!

**Questions:**

- In equation (17), should you be using $b'$ or $b$? It seems more natural to use $b$, and would be confusing to use $b'$, which already includes some scaling modifications.
- For equations 19 and 20, isn't this equivalent to $g(m) = (1-\gamma(r(d))) * (1/s) + \gamma(r(d))$, and $h(\theta_d) = \theta_d$? Perhaps this would be simpler, to make this look more like the position interpolation equations (which only modify $g(m)$), as opposed to the "NTK-aware" equations which only modify $h(\theta_d)$.
- Can you show results of YaRN with and without fine-tuning?
- Can you clarify how the sliding window evaluation method works (Press et al., 2022)? I read the reference but was still confused, so adding a simple explanation in the paper would be very helpful. When reporting on performance with evaluation context window size $N$, does this mean you measure perplexity on predicting token N+1? Or you measure the average perplexity across all $N$ tokens (so each token on average sees N/2 context)?
- When you say in your plots "Yarn-Llama-2-7b-64k", this means you used s=16? What fine-tuning data and sequence lengths did you use for both s=16 and s=32?
- Are the only results in your paper that use dynamic scaling those in Figure 5 of appendix B.4?
- Why do you compare with Code-Llama, which is specialized for code?
- Do you have intuition for why temperature scaling is necessary? Does it improve performance with regular RoPE embeddings, for short documents? Or just for long documents when using YaRN? Can you add more explanation regarding the intuition/reasons for this method, and how it performs outside the context of YaRN?

---

> ### Author Response · Authors · 2023-11-21
>
> ## On the Weaknesses of the Paper
>
> Thank you for your in-depth review of our paper! Thanks to your suggestions, we will shorten the NTK-aware interpolation section and provide a diagram in order to link the four methods outlined in the paper. We will also move a significant portion of the background section to the appendix.
>
> In the case of experimental results, we do agree that it is currently not thorough enough. We would also like to point out that even if YaRN is much more efficient than previous methods, fine-tuning these models and running evaluations at extreme context lengths require significant amounts of computing resources. We are currently finishing the training run for four new LLaMA 7B models using the same training hyperparameters and data, each extended using a different interpolation method (PI, NTK-aware, NTK-by-parts, YaRN). We will be adding a table including all four, and compare their PPL before and after fine-tuning, and also provide PPL for when Dynamic Scaling is enabled/disabled. We will also be comparing these four models on passkey and standard benchmarks in order to establish a better baseline.
>
> For the comparison against other positional embeddings such as T5, ALiBi, absolute, etc. The hard truth is that we simply do not have the required resources in order to perform such extensive tests. In order to compare them fairly to RoPE and avoid confounding factors such as different preexisting base models being pre-trained on different datasets (eg. LLaMA's RoPE vs BLOOM's ALiBi), this experiment would each require us to pretrain and then fine-tune many models from scratch using each of the positional embeddings, and we simply do not have the resources to pretrain a model from scratch. For now, our work will only focus on RoPE, which is the most popular and most used positional embedding of all public models.
>
> We will try to provide better tables and better plots, and also move the current comparisons against Together and Code Llama to the appendix. Those were chosen as there were no other publicly available long context models that we could compare against our 64k and 128k context models. The new PPL comparisons will include an extensive ablation study comparing all different methods and different scaling factors before and after fine-tuning.
>
> ## Answers to Questions
>
> - For equation 17, it is a typo, we will fix it.
> - Yes, both interpretations of the NTK-by-parts equations for eq 19 and 20 are equivalent, we can change it but the original NTK-by-parts equation was derived to what is included in the paper.
> - We will provide that comparison in the next draft.
> - The sliding window evaluation is basically the standard perplexity evaluation, but we slide the window across the entire context of the document in order to evaluate its average perplexity across the entire document. For non-sliding window evaluation, the document is usually truncated to the context size from the beginning, but we lose the perplexity contribution of the later tokens in the document. By sliding the context window across the document and iteratively truncating and averaging different parts of the document, we can take in account the perplexity of the entire document, even if the context size is shorter.
> - Both s=16 and s=32 models were trained using a dataset of 64k context. The s=32 model extrapolates to 128k context without any issues even if it had only seen 64k context during training.
> - We will provide more dynamic scaling results in the next draft.
> - We will compare against a better baseline in the next draft, we did not have any other publicly available models to compare against and we initially could not waste any compute in training an inferior PI model at 64k or 128k.
> - This result is only backed by empirical evidence, but we are trying to figure out why temperature scaling significantly improves PPL in the case of RoPE. We currently do not have a satisfactory answer to this question.

---

> > ### Comment · Reviewer_x5Kh · 2023-12-04
> >
> > Thank you very much to the authors for their thoughtful response to my review. After reading this response and the other reviews, and seeing the updates to the paper, I have updated my score to "marginal accept". Thank you very much to the authors for the improvements they have made to the paper.

---

### Meta-Review · Area_Chair_Merr · 2023-12-06

**Metareview:**

The paper presents YaRN (Yet another RoPE extensioN method), a compute-efficient method to extend the context window of large language models (LLMs) trained with Rotary Position Embeddings (RoPE). YaRN combines attention scaling, "NTK-by-parts" interpolation, and dynamic scaling to achieve state-of-the-art performance in context window extensions. It is compatible with libraries that modify the attention mechanism, such as Flash Attention, and allows for efficient extrapolation with fine-tuning on shorter datasets.

The reviewers generally appreciated the novelty and effectiveness of the YaRN method, noting its potential to extend the context window of LLMs more efficiently than previous methods. However, some reviewers found the paper difficult to follow and suggested that the method could be presented in a simpler manner. They also suggested more comprehensive comparisons with other length extending methods and more detailed ablation studies. Despite these concerns, the reviewers generally agreed that the paper makes a valuable contribution to the field.

**Justification For Why Not Higher Score:**

The major concern of this work lies in the clarity of the presentation and the comprehensiveness of the experimental results. Some reviewers found the paper difficult to follow and suggested that the method could be presented in a simpler and more direct manner. They also called for more comprehensive comparisons with other length extending methods and more detailed ablation studies.

**Justification For Why Not Lower Score:**

See meta review

---

### Decision · Program_Chairs · 2024-01-16

Accept (poster)